# Solution X-Ray Absorption Spectroscopy (XAS) for Analysis of Catalytically Active Species in Reactions with Ethylene by Homogeneous (Imido)vanadium(V) Complexes—Al Cocatalyst Systems

**Kotohiro Nomura** 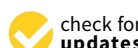

Department of Chemistry, Graduate School of Science, Tokyo Metropolitan University, 1-1 Minami Osawa, Tokyo 192-0397, Japan; ktnomura@tmu.ac.jp; Tel.: +81-42-677-2547

**Abstract:** Solution V K-edge XANES (X-ray absorption near edge structure) and EXAFS (extended X-ray absorption fine structure) analysis of vanadium(V) complexes containing both imido ligands and anionic ancillary donor ligands (L) of type, $V(NR)(L)X_2$ (R = Ar, Ad (1-adamantyl); Ar = $2,6-Me_2C_6H_3$; X = Cl, Me, L = $2-(ArNCH_2)C_5H_4N$, OAr, WCA-NHC, and 2-(2′-benzimidazolyl)pyridine; WCA-NHC = anionic NHCs containing weak coordinating $B(C_6F_5)_3$), which catalyze ethylene dimerization and/or polymerization in the presence of Al cocatalysts, has been explored. Different catalytically actives species with different oxidation states were formed depending upon the Al cocatalyst (MAO, $Me_2AlCl$, $Al^iBu_3$, etc.) and the anionic ancillary donor ligand employed. The method is useful for obtainment of the direct information of the active species (oxidation state, basic framework around the centered metal) in solution, and for better understanding in catalysis mechanism and organometallic as well as coordination chemistry.

**Keywords:** vanadium catalyst; catalysis mechanism; polymerization; dimerization; XANES; EXAFS; Al cocatalyst; homogeneous catalysis; vanadium complex; active species

## 1. Introduction

In homogeneous molecular catalysis, direct analysis of the active site in catalysis solution has been considered as an important subject for better understanding not only in structural and electronic nature of the reaction sphere in situ, but also in basic catalysis mechanism, and for better catalyst design. However, as described below, researchers in the field of molecular catalysis have a strong desire to obtain the information on the actual reaction sphere in solution. Nuclear magnetic resonance (NMR) is the principal technique for identification of organic/inorganic, organometallic compounds in (partially) deuterated solvent [1–4]. X-ray crystallographic analysis is the conventional method for obtainment of information in the single crystal of the intermediate(s) in the proposed catalysis cycle, which is (are) generally isolated by certain stabilization from the ideal reaction solution [1–4]. On the basis of these analysis results and results in some reaction chemistry with the proposed model complexes (including direct use of the isolated species in the catalytic reaction) as well as computational analysis, we usually propose a catalysis mechanism. Moreover, the approach by electron spin resonance (ESR) spectroscopy not by NMR spectroscopy has been employed for the paramagnetic compounds [5,6], but the method lacks the quantitative analysis in addition to a possibility of formation of "ESR silent" species. Moreover, it is difficult to obtain (clear) structural information in solution by these (NMR and ESR) methods. In some cases, isolated intermediates from the model reaction are too stable (due to certain stabilization for the isolation or isolated as a dormant and the side product in the catalysis cycle) and these species often show low (or negligible) activity compared to the originals. Moreover,

the structural information (bond distances and angles) provided by the X-ray crystallography should be somewhat different from those in the homogeneous solution. Therefore, development of a new method for obtainment of structural and electronic information of the catalytically active species in solution has been a topic of particular interests in the field of homogeneous catalysis as well as organometallic chemistry.

Synchrotron XAS (X-ray absorption spectroscopy) is the method for analysis of the oxidation state and the geometry through XANES (X-ray absorption near edge structure) analysis and of their coordination atoms to the metal center through EXAFS (extended X-ray absorption fine structure) analysis. These are one of the most popular methods for study in heterogeneous catalysis [7–13], and are also considered as the effective methods for study in homogeneous (molecular) catalysis recently [14–26]. We recently introduced that the method is highly useful for analysis of the vanadium catalyst solution, especially for ESR silent paramagnetic species, which cannot be observed by both NMR and ESR spectra [19–23,25,26].

Design and synthesis of vanadium complex catalysts for efficient olefin polymerization, oligomerization has been considered as a promising subject in the filed of catalysis, organometallic chemistry, and of polymer chemistry [27–32]. This is because of unique characteristics demonstrated by classical Ziegler-type vanadium catalyst systems (e.g., $VOCl_3$, $VCl_4$, $VCl_3$-$AlBr_3$, $AlCl_3$-$AlPh_3$, $Al^iBu_3$, and $SnPh_4$), which exhibit significant initial reactivity toward olefins (affording ultrahigh molecular weight polymers with narrow distributions) [27–38]. However, these polymerizations were generally conducted under deep cool conditions (below 0 °C, even in the commercial production for synthetic rubbers, EPDM) and their rapid catalyst deactivation causes poor overall productivities. The short catalyst life could be partly overcome upon addition of ethyl trichloroacetate (ETA) [39], due to an assumption that inactive vanadium(II) species were re-oxidized by ETA [39]. Therefore, most of ethylene polymerizations using vanadium catalyst systems were conducted in the presence of large excess ETA [33–38]. This was also because vanadium(III) species were assumed to be the active species based on an electron spin resonance (ESR) spectra and titration study [40–43]. Although ESR spectroscopy is effective for analysis of paramagnetic compounds [5,6,44–48], this cannot be applied to "ESR silent" vanadium(III) species and lacks their quantitative analysis as well as for obtainment the structural information.

Recently, we demonstrated that the XAS should be useful for analysis of the vanadium catalyst solution especially containing ESR silent paramagnetic species. Although, as described above, no or negligible resonances were observed from both the ${}^{51}$V NMR and the ESR spectra, existence of vanadium(III) species showed apparent different edge absorption [20–23], as described below. Moreover, the EXAFS analysis can provide information concerning the (number and kind of) atoms coordinated to vanadium, as described above. The XANES analysis should be useful for exploring the oxidation state of the formed species upon addition of Al cocatalysts (and monomers), etc.

In this reviewing feature article, we first introduce the XANES spectra of some vanadium complexes (compounds) with different oxidation states and geometries for explanation of basics [25,26,49–52], and then introduce our recent results for analysis of the catalytically active species in vanadium catalyzed ethylene polymerization and dimerization through XAS (XANES and EXAFS) analysis. Through these studies, we wish to emphasize the promising potential of these methods for direct analysis of the catalytically active species in the molecular level [19–23,25,26].

## 2. V-K Edge XANES (X-ray Absorption Near Edge Structure) and EXAFS (Extended X-ray Absorption Fine Structure) Spectra of Vanadium Complexes: Effect of Oxidation State, Geometries toward the Pre-Edge Peak Intensity, and the Edge Absorption in the XANES Spectra

It has been known that the pre-edge peak intensity and the edge absorption (peak) in the V-K edge XANES spectra are influenced by the oxidation state and basic geometry around vanadium [49–52]. This is because that transition of a 1s electron to 3d orbital is the electric quadrupole transition (giving weak pre-edge absorptions), and an intense pre-edge peak is assigned to an electric dipole transition to

the p-character in the d–p hybridized orbital; simple estimation of the pre-edge peak intensity could be thus possible from the character tables of group theory [50,51]. For example, in general, an intensity in the pre-edge absorption in a compound with $T_d$ symmetry shows higher than that in a compound with $O_h$ symmetry, due to difference in the possibility of the orbital hybridization. No irreducible representations in both d and p orbitals are present in the $O_h$ point group, whereas a p–d hybridized orbital could be formed with the presence of irreducible representation ($p_x$, $p_y$, and $p_z$ and $d_{xy}$, $d_{yz}$, and $d_{zx}$ orbitals) in $T_d$ symmetry [50,51].

Figure 1a shows V K-edge XANES spectra (5.46 keV, in solid state at 25 °C) for vanadium oxides ($V_2O_3$, $VO_2$, and $V_2O_5$), which are often used for estimation of the oxidation state (by peak position (inflection point) in the edge absorption). $V_2O_3$ and $VO_2$ exhibit a distorted octahedral geometry; $V_2O_3$ has a corundum structures (with two different V–O distances) and $VO_2$ is monoclinic (is distorted from of rutile) with short V–O distance. Geometry of $V_2O_5$ is distorted square pyramidal with apparent different distance between apex- and basal-oxygens. Their pre-edge peaks observed at 5468.4 and 5469.8 eV ($V_2O_3$), 5470.0 eV ($VO_2$), and 5470.8 eV in $V_2O_5$ are, as described above, generally considered as due to a transition from 1s to 3d + 4p [50,51]. It has been known that the pre-edge XANES peak characteristics of 3d transition metals are affected by the coordination sphere symmetry, coordination number, and number of d-electrons [50,51].

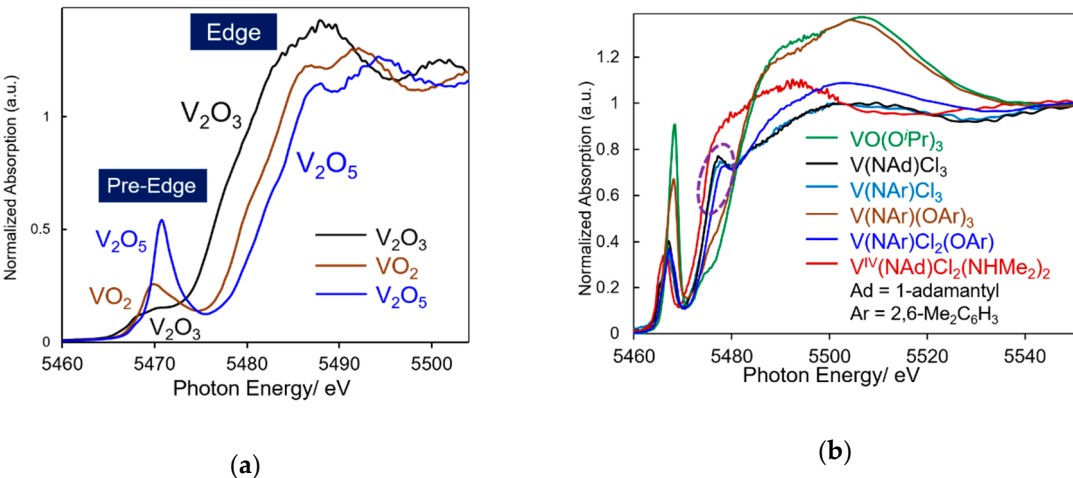

(**a**)             (**b**)

**Figure 1.** V K-edge X-ray absorption near edge structure (XANES) spectra (5.46 keV, through the use of synchrotron radiation at SPring-8, BL01B1 beamline) for: (**a**) vanadium oxides ($V_2O_3$, $VO_2$, and $V_2O_5$) in solid state (at 25 °C); (**b**) selected four or five coordinate vanadium complexes (in toluene at 25 °C).

Figure 1b shows XANES spectra of a series of four coordinate vanadium(V) complexes (VO(O$^i$Pr)$_3$, V(NAd)Cl$_3$ [53], V(NAr)Cl$_3$ [54], V(NAr)(OAr)$_3$ [55], and V(NAr)Cl$_2$(OAr) [56]; Ad = 1-adamantyl, and Ar = 2,6-Me$_2$C$_6$H$_3$) in toluene at 25 °C, and the spectrum of V$^{IV}$(NAd)Cl$_2$(NHMe$_2$)$_2$ [45] is shown for comparison. These vanadium(V) complexes show similar pre-edge peak positions (5468.4 eV in VO(O$^i$Pr)$_3$, 5465.6 eV (and 5467.1 eV) in V(NAd)Cl$_3$ [19], at 5465.0 eV (and 5466.8 eV) in V(NAr)Cl$_3$ [19], 5468.1 eV in V(NAr)(OAr)$_3$ [23], and 5466.8 (and 5465.3) eV in V(NAr)Cl$_2$(OAr) [21]), which are considered as due to a transition from 1s to 3d + 4p [25,26,49,50]; the curve-fitting of XANES spectrum of V(NAd)Cl$_3$ is in good agreement with that estimated by the DFT (density functional theory) calculation [19]. Strong pre-edge peak intensities observed especially in VO(O$^i$Pr)$_3$ and V(NAr)(OAr)$_3$ are due to their tetrahedral ($T_d$) geometry around vanadium (formation of d–p hybridized orbital through 1s–3d transition), as described above [25,26,49,50]. In addition to the pre-edge peaks, the XANES spectra in V(NAd)Cl$_3$, V(NAr)Cl$_3$, and V(NAr)Cl$_2$(OAr) showed a "shoulder-edge absorption" (marked in circle at 5476.7, 5477.6, and 5475.7 eV, respectively), which are considered to be ascribed to an absorption of the V–Cl bond [25,26,57].

The spectrum for five coordinate vanadium(IV) complex, V(NAd)Cl$_2$(NHMe$_2$)$_2$, showed a rather broad pre-edge peak at 5466.4 eV [20], which is close to that in vanadium(V) complex, V(NAd)Cl$_3$ (5465.6 eV (and 5467.1 eV), Figure 1b), but the edge absorption shifted to low energy compared to those in the (imido)vanadium(V) di-, trichlorides. These results thus suggest that the oxidation state can be estimated by the edge absorptions among a series of (imido)vanadium complexes.

Figure 2a shows V K-edge XANES spectra (in toluene at 25 °C) for four coordinate V(NAr)Cl$_3$, V(NAr)Cl$_2$(WCA-NHC) [58], V(NAr)Cl$_2$(N=C$^t$Bu$_2$) [59], and six coordinate V(NAr)Cl$_2$(N=C$^t$Bu$_2$)(dmpe) (dmpe = 1,2-bis(dimethylphosphino)ethane) [59]. These four coordinate complexes showed similar pre-edge peaks with rather strong intensities due to their distorted tetrahedral ($T_d$) geometry around vanadium; the pre-edge intensity of six coordinate V(NAr)Cl$_2$(N=C$^t$Bu$_2$)(dmpe) (with $O_h$ geometry) was rather weak compared to their four coordinate tetrahedral complexes, as explained above. In all complexes, as observed in Figure 1b, unique absorptions called shoulder-edge ascribed to V–Cl absorptions were observed. Similarly in Figure 2b, both V(NAd)Cl$_3$ and V(NAd)Cl$_2$[2-(ArCH$_2$)C$_5$H$_4$N] [60] showed shoulder-edge absorptions, whereas as no such shoulder absorption was observed in V(NAd)Me$_2$[2-(ArCH$_2$)C$_5$H$_4$N]; clear two pre-edge absorptions in V(NAd)Cl$_2$[2-(ArCH$_2$)C$_5$H$_4$N] are due to the presence of two major d–p hybridizations (transitions) [19].

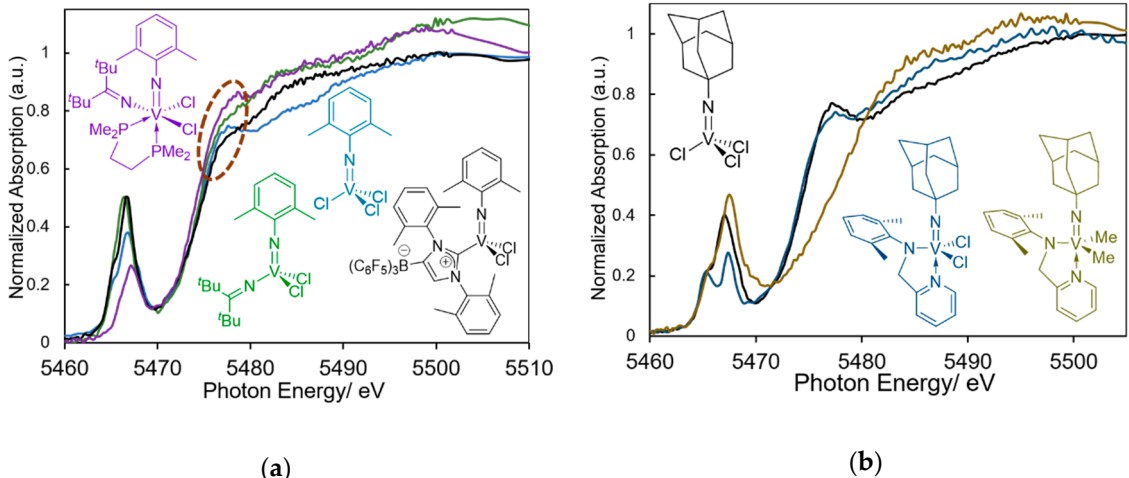

(**a**)            (**b**)

**Figure 2.** Solution V K-edge XANES spectra (5.46 keV, in toluene at 25 °C) for: (**a**) V(NAr)Cl$_3$, V(NAr)Cl$_2$(WCA-NHC), V(NAr)Cl$_2$(N=C$^t$Bu$_2$), and V(NAr)Cl$_2$(N=C$^t$Bu$_2$)(dmpe) and (**b**) V(NAd)Cl$_3$ and VN(Ad)X$_2$[2-(ArNCH$_2$)C$_5$H$_4$N] (X = Cl, Me).

Figure 3 shows V K-edge EXAFS oscillations (left) and FT-EXAFS spectra (right) and their simulation curves (on the basis of X-ray crystallographic data) for V(NAd)Cl$_2$[2-(ArNCH$_2$)C$_5$H$_4$N] (in toluene at 25 °C) [19]. The spectrum showed good agreement with the fitting curves by simulations on the basis of X ray crystallographic analysis data and of DFT calculation [19]. Two peaks (at 1.2 Å and 1.9 Å in Figure 3, right) observed in the spectrum were determined as V–N and V–Cl bonds, respectively, by comparison of backscattering amplitude obtained from the spectrum by reverse Fourier transform analysis with those by the FEFF calculation (an automated program for ab initio multiple scattering calculations of XAFS) [19]. The EXAFS analysis data shows that one short V–N (imido) bond (1.66(3) Å), and two V–Cl bonds (2.274(3) Å) were close to those observed by the X-ray crystallographic analysis (1.6517(12) Å, 2.2677(3), and 2.2709(4) Å, respectively), whereas the presence of an additional one of two V–N bonds (1.88(2) Å) corresponding to the (2-anilidemethyl)pyridine ligand are somewhat different (1.8580(12) and 2.2241(11) Å) from those by the X-ray crystallography.

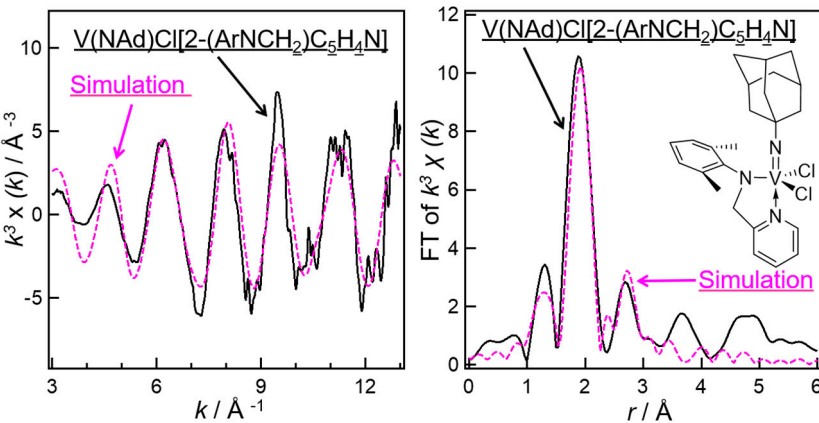

**Figure 3.** V K-edge extended X-ray absorption fine structure (EXAFS) oscillations (**left**) and FT-EXAFS spectra (**right**) and their simulation curves (on the basis of X-ray crystallographic data) for V(NAd)Cl$_2$[2-(ArNCH$_2$)C$_5$H$_4$N] (in toluene at 25 °C, Ad = 1-adamantyl, and Ar = 2,6-Me$_2$C$_6$H$_3$) [19]. In this simulation, the contributions from the neighbor atoms within 3.3 Å distance from V atom were considered and 0.0036 Å$^2$ of the Debye–Waller factor was applied.

## 3. Solution XANES Analysis for Exploring Oxidation State of the Catalytically Active Species in Ethylene Dimerization/Polymerization Using V(NAd)X$_2$[2-(ArNCH$_2$)C$_5$H$_4$N] (X = Cl, Me)—Al Cocatalyst Systems

Ethylene oligomerization for production of linear α-olefins has been one of the key reactions in the chemical industry [31,61–77], and nickel complex catalysts containing a chelate P−O ligand has been known as the shell higher olefin process (SHOP) [61–63], ethylene dimerization using Ti(OBu)$_4$–AlEt$_3$ [67,68], and ethylene trimerization using chromium [78] or titanium [79] catalysts have been known. Development of the highly active and selective transition metal catalysts, such as nickel [61–63,66,73], iron and cobalt [72,74–76], and chromium [69–71,78], has thus been the attractive subject in the field of catalysis and organometallic chemistry [31,61–77]. 1-Butene is the important liner α-olefin used not only as a comonomer in the production of linear low density polyethylene (LLDPE), but also as the principle starting material for manufacturing a range of valuable synthetic intermediates. As described above, the catalyst system consisting of Ti(OBu)$_4$, AlEt$_3$ and additives, which showed moderate catalytic activity (turnover frequency—TOF = 21,200 h$^{-1}$), has been known as the practical catalyst for ethylene dimerization [67,68]. As shown in Scheme 1, ethylene oligomerization proceeds via olefin insertion (metal-hydride or metal-alkyl) or metallacycle mechanism.

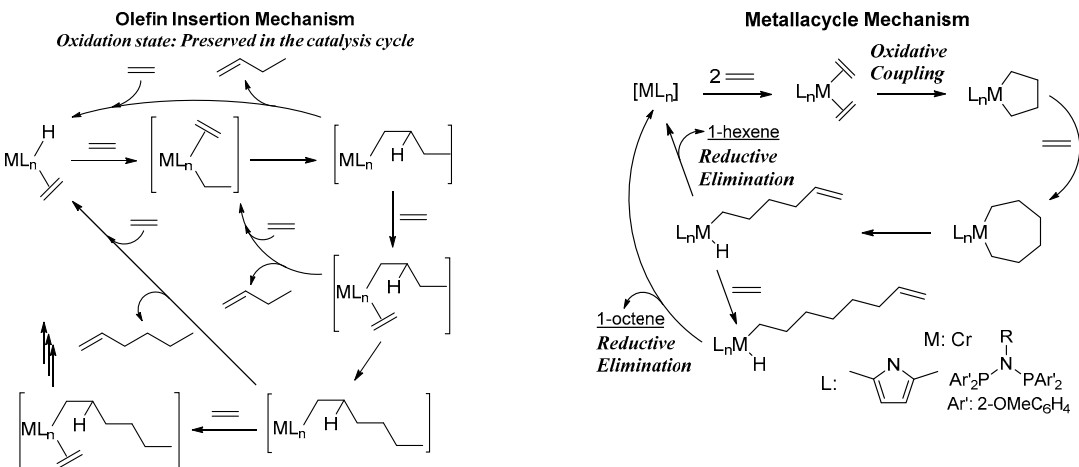

**Scheme 1.** Two mechanisms, olefin insertion (through metal-hydride or metal-alkyl intermediate, (**left**)) or metallacycle intermediate (**right**), considered in ethylene oligomerization.

The selective ethylene trimerization (and tetramerization) affording 1-hexane (and 1-octene) has been proposed via the metallacycle (metallacycloheptane) intermediate, whereas nickel catalyzed ethylene oligomerization exemplified as SHOP process proceeds via olefin insertion mechanism (affording a mixture of oligomers with certain distribution, called the Schulz–Flory distribution). In these mechanisms, oxidation state has been preserved in the olefin insertion mechanism (the reaction should be the first order dependent on the ethylene pressure), whereas oxidative addition (change in the oxidation state) involves in the metallacycle mechanism (the reaction rate should the second order dependent upon the ethylene pressure).

(Imido)vanadium(V) complexes that contain (2-anilidemethyl)pyridine or 8-anilide-5,6,7-tri-hydroquinoline ligand of type, $V(NAd)X_2[2-(ArNCH_2)C_5H_4N]$ [X = Cl (**1**), Me (**2**); Ar = 2,6-$Me_2C_6H_4$] [19,60], or $V(NAd)Cl_2[8-ArN(C_9H_{10}N)]$ (**3**) [80], have been known to be the efficient catalyst precursors for selective ethylene dimerization in the presence of methylaluminoxane (MAO) cocatalyst. As shown in Table 1 [60,80], the activities by **1,2**–MAO catalyst (TOF = 1,830,000–3,460,000 $h^{-1}$ (508–961 $s^{-1}$); TOF = (ethylene reacted in mol)/(mol-V loaded)·(time)) [60] showed much higher than that by $Ti(OBu)_4$–$AlEt_3$ based catalyst systems (TOF = 21,200 $h^{-1}$ (5.89 $s^{-1}$)) [67,68]; complex **3** showed the higher activity (ex. TOF: 9,600,000 $h^{-1}$ (2670 $s^{-1}$)) affording 1-butene as the major product (95.0%–99.4%) [80]. The dimethyl analogue (**2**) showed similar catalyst performance to **1**, strongly suggesting that similar catalytically active species play the role in this catalysis [60].

**Table 1.** Reaction with ethylene catalyzed by $V(NAd)X_2[2-(ArNCH_2)C_5H_4N]$ (X = Cl (**1**), Me (**2**); Ad = 1-adamantyl, and Ar = 2,6-$Me_2C_6H_3$), $V(NAd)Cl_2[8-ArN(C_9H_{10}N)]$ (**3**) in the presence of the Al cocatalyst [60,80,81][a].

| Cat. | Al Cocat. | Al/V | Oligomer | | | PE | |
|---|---|---|---|---|---|---|---|
| (µmol) | | Molar Ratio | Activity [b] | TOF [c]/$h^{-1}$ ($s^{-1}$) | C4′ [d]/% | Activity [b] | $M_\eta$ [e] $\times 10^{-6}$ |
| **1** (0.5) | MAO | 500 | 50,100 | 1,830,000 (508) | 92.5 | – | – |
| **1** (0.1) | MAO | 1500 | 76,500 | 2,730,000 (758) | 97.0 | – | – |
| **1** (5.0) | $Me_2AlCl$ | 200 | trace | – | – | 704 | 5.92 |
| **1** (5.0) | $Et_2AlCl$ | 100 | trace | – | – | 137 | 6.76 |
| **2** (0.5) | MAO | 500 | 98,800 | 3,460,000 (961) | 91.0 | – | – |
| **2** (0.5) | MAO | 1500 | 82,300 | 2,880,000 (800) | 87.9 | – | – |
| **3** (0.1) | MAO | 3000 | 205,400 | 7,190,000 (2000) | 97.7 | – | – |
| **3** (0.1) | MAO | 4000 | 274,000 | 9,600,000 (2670) | 97.7 | – | – |

[a] Conditions: ethylene 8 atm, toluene 30 mL, d-MAO white solid, 25 °C (or 0 °C with $Me_2AlCl$, $Et_2AlCl$), 10 min. [b] Activity in kg-ethylene reacted/mol-V·h. [c] TOF (turnover frequency) = (molar amount of ethylene reacted)/(mol-V·h). [d] By GC analysis vs. internal standard. [e] Molecular weight by viscosity.

The TOF value by **1** showed the first order dependence toward the ethylene pressure in the presence of modified MAO (MMAO, methyl isobutyl aluminoxane) [81], suggesting that the reaction proceeds via insertion mechanism (with the metal-hydride or metal-alkyl species, Scheme 1 right). In contrast, **1** afforded linear polyethylene with ultrahigh molecular weight in the presence of $Me_2AlCl$, and $Et_2AlCl$ (in place of MAO) [81]; the results also suggest the insertion mechanism with cationic metal-alkyl species. Reactions of **1** with 10.0 equiv of MAO, MMAO, $Et_2AlCl$, and $Me_2AlCl$ led to formation of the other diamagnetic vanadium(V) species (observed as different resonances from **1**) in the $^{51}V$ NMR spectra (in $C_6D_6$ and toluene-$d_8$), whereas treatment of **1** in toluene with MMAO, $Et_2AlCl$ (100 equiv) led to formation of paramagnetic species (observed as a disappearance of resonances in the ESR spectra) [60,81]. Based on these results, as explained in Scheme 2, it was proposed that the observed difference could be assumed as due to the formation of two catalyst species containing different counter anions; isolated cationic (for dimerization) or associated cationic species (for polymerization), which could facilitate or disturb the β-hydrogen elimination [81]. It was however pointed out that these results do not entirely exclude a possibility of formation of ESR silent vanadium(IV) dimers coupled antiferromagnetically or vanadium(III), proposed in the classical Ziegler-type catalyst systems.

Therefore, synchrotron K-edge XAS analysis was focused to obtain the information of oxidation state in the catalyst solution.

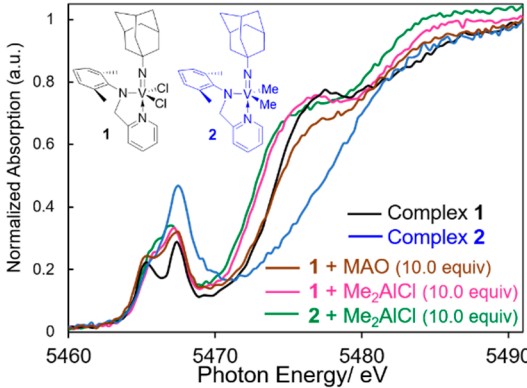

**Scheme 2.** Ethylene dimerization yielding 1-butene and ethylene polymerization using V(NAd)X$_2$[2-(ArNCH$_2$)C$_5$H$_4$N] (X = Cl (**1**), Me (**2**)] or V(NAd)Cl$_2$[8-ArN(C$_9$H$_{10}$N)) in the presence of Al cocatalysts, including proposed (isolated, associated) catalytically active species.

As described above (in Section 2), it has been known that the pre-edge peak intensity and the edge absorption (peak) in the V-K edge XANES spectra are influenced by the oxidation state and basic geometry around vanadium [49–52]. Figure 4 shows the XANES spectra in toluene containing the dichloride complex (**1**) and MAO (10.0 equiv) at 25 °C, and the spectrum for the dimethyl complex (**2**) is also shown for comparison. Importantly, a significant change in the pre-edge peak positions and intensities caused by addition of MAO (5465.1 and 5467.6 eV (pre-edge)) was not observed, whereas decrease in the shoulder edge intensity at 5477.8 eV (ascribed to the V–Cl bond) was observed upon addition of MAO [60]. Decrease in the absorption was corresponded to a decrease in V–Cl peak intensity and the coordination number in the FT-EXAFS spectrum. These thus clearly suggest that the coordinated Cl ligand in **1** was cleaved by reacting with MAO without reacting with any V–N bonds. On the basis of (i) $^{51}$V NMR and ESR spectra [81], (ii) isolation of the dimethyl (and cationic) complexes and reaction chemistry (including effect of Al cocatalysts, ethylene pressure dependence) [19,81], and (iii) the XANES and EXAFS analysis [19], it is concluded that cationic vanadium(V) alkyl species play a role as the catalytically active species in the ethylene dimerization [59].

**Figure 4.** V K-edge XANES spectra (in toluene at 25 °C, 5.46 keV, through the use of synchrotron radiation at SPring-8, BL01B1 beamline) for V(NAd)X$_2$[2-ArNCH$_2$C$_5$H$_4$N] (X = Cl (**1**), Me (**2**)) with 10 equiv of MAO or Me$_2$AlCl [19].

As also shown in Figure 4, treatment of **1** with 10 equiv of Me$_2$AlCl led to a slight low energy shift in the shoulder-edge in **1** without significant changes in the pre-edge peaks (5476.4 eV). The similar spectral change was observed in the treatment of the dimethyl complex (**2**) with 10 equiv of Me$_2$AlCl; the shoulder-edge absorption at 5476.4 eV clearly suggests formation of V–Cl bond (in high certainty

through Al), and the results are in good agreement with those observed in the $^{51}$V NMR spectra and the reaction chemistry shown in Scheme 2 [19,81]. It is thus concluded that (2-anilidomethyl)pyridine ligand plays a role to stabilize the oxidation state in reaction with Al cocatalysts (MAO, Me$_2$AlCl, and Et$_2$AlCl) in toluene.

## 4. Solution XANES Analysis for Exploring the Oxidation State of Catalytically Active Species in Ethylene Polymerization Using (Imido)vanadium(V) Dichloride Complex Catalysts Containing Anionic Ancillary Donor Ligands in the Presence of Al Cocatalysts

(Imido)vanadium(V) complexes containing monodentate anionic ancillary donor ligand (Y) of type, V(NAr)Cl$_2$(Y) (Y = phenoxide (**4**) [56,82,83], iminoimidazolide (**5**) [84], and iminoimidazolidide (**6**) [84]), shown in Scheme 3, exhibit (significant) catalytic activities for ethylene polymerization and the copolymerization with norbornene (NBE) in the presence of MAO or Me$_2$AlCl, and Et$_2$AlCl as a cocatalyst. Selected results in ethylene polymerization by **4–6**, and **8** in the presence of Al cocatalysts are summarized in Table 2. In these complexes, the activities in the presence of Me$_2$AlCl or Et$_2$AlCl (runs 2–4,10,11,13,14) were higher than those in the presence of MAO (runs 1, 9, and 12), affording ultrahigh molecular weight linear polymers. The activity was affected by the Al/V molar ratio, ethylene pressure, and the temperature; these (co)polymerizations in the presence of halogenated Al alkyl cocatalysts were conducted at 0 °C due to significant decrease in the activity at 25 °C probably by the catalyst decomposition (run 10 vs. 11, and run 13 vs. 14, Table 2). Remarkable decrease in the activity was observed in the polymerization by **4** upon addition of Cl$_3$CCO$_2$Et (ETA, run 5), which are widely used as activators in (especially classical Ziegler type) vanadium catalysts [33–38]; use of halogenated Al alkyl should be prerequisite for exhibiting the high activity by **4** (runs 2–4, and 6–8).

**Scheme 3.** Selected (imido)vanadium(V) dichloride complex catalysts for ethylene polymerization.

The 2-(2′-benzimidazolyl)pyridine analogue (**8**) also showed the notable activities in the presence of Me$_2$AlCl cocatalyst, whereas the resultant products with ethylene were a mixture of polymer and oligomers if the reaction by **8** was conducted in the presence of MAO [22]. The (2-anilidemethyl)pyridine analogue containing arylimido ligand, V(NAr)Cl$_2$[(2-ArCH$_2$)C$_5$H$_4$N] (**7**), also showed the activity for ethylene polymerization, whereas the adamantylimido analogues showed remarkable catalytic activities for ethylene dimerization in the presence of MAO, as described above (Scheme 2, Table 1) [60,81,85]. As shown in Table 2, the activity by **8** increased upon addition of ETA [22], and this should be a unique contrast to that observed in the phenoxide analogue (**4**) [83].

**Table 2.** Ethylene polymerization by V(NAr)Cl$_2$(L) (L = OAr (**4**), 1,3-Ar$_2$(CHN)$_2$C=N (**5**), and 1,3-Ar$_2$(CH$_2$N)$_2$C=N (**6**)) and V(NAd)Cl$_2$[2-(2'-benzimidazolyl)-6-methylpyridine] (**8**), in the presence of the Al cocatalyst [22,56,82–84] [a].

| Run | Cat. (µmol) | Al Cocat. | Al/V [b] | ETA [c] | Temp./°C | Time/min | Activity [d] |
|---|---|---|---|---|---|---|---|
| 1 | **4** (1.0) | MAO | 2500 | – | 25 | 10 | 2930 |
| 2 | **4** (0.05) | Me$_2$AlCl | 5000 | – | 0 | 10 | 27,500 |
| 3 | **4** (0.05) | Et$_2$AlCl | 5000 | – | 0 | 10 | 11,700 |
| 4 | **4** (0.05) | Et$_2$AlCl | 5000 | – | 0 | 10 | 11,400 |
| 5 | **4** (0.05) | Et$_2$AlCl | 5000 | 10 | 0 | 10 | 1080 |
| 6 | **4** (0.05) | $^i$Bu$_2$AlCl | 5000 | – | 0 | 10 | 52,000 |
| 7 | **4** (1.0) | Et$_2$Al(OEt) | 500 | – | 0 | 10 | trace |
| 8 | **4** (1.0) | $^i$Bu$_3$Al | 500 | – | 0 | 10 | trace |
| 9 | **5** (2.0) | MAO | 1000 | – | 25 | 10 | 507 |
| 10 | **5** (0.04) | Et$_2$AlCl | 2000 | – | 0 | 10 | 38,300 |
| 11 | **5** (0.04) | Et$_2$AlCl | 2000 | – | 25 | 10 | 7650 |
| 12 | **6** (2.0) | MAO | 1000 | – | 25 | 10 | 627 |
| 13 | **6** (0.04) | Et$_2$AlCl | 1000 | – | 0 | 10 | 32,000 |
| 14 | **6** (0.04) | Et$_2$AlCl | 1000 | – | 25 | 10 | 6000 |
| 15 | **8** (0.05) | Me$_2$AlCl | 5000 | – | 25 | 6 | 21,700 |
| 16 | **8** (0.05) | Me$_2$AlCl | 10,000 | – | 25 | 6 | 55,700 |
| 17 | **8** (0.05) | Me$_2$AlCl | 10,000 | 50 | 25 | 6 | 46,700 |
| 18 | **8** (0.05) | Me$_2$AlCl | 10,000 | 100 | 25 | 6 | 82,000 |

[a] Conditions: ethylene 8 atm, toluene 30 mL. [b] Molar ratio to V. [c] Co-presence of CCl$_3$CO$_2$Et (ETA, molar ratio to V). [d] Activity in kg-PE/mol-V·h.

The WCA-NHC analogues (**9**), consisting of anionic *N*-heterocyclic carbenes (NHCs) containing a weakly coordinating tris(pentafluorophenyl)borane moiety, showed high catalytic activities for ethylene polymerization in the presence of Al$^i$Bu$_3$ [23,58], whereas Al$^i$Bu$_3$ is in most case inactive cocatalyst for the metal catalyzed olefin polymerization. As summarized in Table 3, the diisopropylphenyl analogue (**9b**) showed the higher activities than the dimethylphenyl analogue (**9a**), and the activities in the presence of Al$^i$Bu$_3$ cocatalyst showed higher than those in the presence of MAO (and AlEt$_3$, and Et$_2$AlCl) cocatalyst. It turned out that the activity by **9b** (19,500 kg-PE/mol-V·h, run 21) is higher than those reported by the other (imido)vanadium(V) complexes in the presence of MAO cocatalyst [23]. Moreover, the **9b**–Al$^i$Bu$_3$ catalyst showed the higher activity (66,000 kg-PE/mol-V·h, run 29) than those not only by the **9b**–MAO catalyst, but also by the other (imido)vanadium(V) complexes in the presence of Me$_2$AlCl or Et$_2$AlCl [23].

Figure 5a shows V K-edge XANES spectra (in toluene at 25 °C) for V(NAr)Cl$_2$(OAr) (**4**), and V(NAr)Cl$_2$(WCA-NHC) (**9b**) upon addition of MAO (10.0 equiv). As described above, the XANES spectra for **4** showed the pre-edge peak (and a shoulder peak) at 5466.8 (and 5465.3) eV, which are similar to those in **9b** (5466.9 (and 5465.1) eV). The peak positions in the spectra of **4** did not change (5466.8 and 5467.0 eV, respectively) upon addition of MAO (10 equiv) with increasing in the pre-edge peak intensity [21,23], with a slight shift of the edge absorptions. The pre-edge peaks in **9b** were shifted slightly at 5464.5 eV with decreasing in intensity of the shoulder-edge peak by addition of MAO (10 equiv) [23]; similar results were observed when **9a** was treated with MAO (10.0 equiv) [20]. Moreover, as shown in Figure 5b the edge absorptions in both **4** and **9b** did not change upon addition of MAO. These results thus suggest that the oxidation state and the basic framework were preserved even by treatment with MAO.

**Table 3.** Ethylene polymerization by V(NAr)Cl$_2$(WCA-NHC) (R' = Me (**9a**), $^i$Pr (**9b**)) in the presence of the Al cocatalyst [23,58] $^a$.

| Run | Cat. (µmol) | Al Cocat. | Al/V $^b$ | Temp./°C | Activity $^c$ | $M_n$ $^d$ $\times 10^{-4}$ | $M_w/M_n$ $^d$ |
|---|---|---|---|---|---|---|---|
| 19 | **9a** (1.0) | MAO | 500 | 25 | 1580 | 1.53 | 1.93 |
| 20 | **9b** (0.2) | MAO | 2000 | 25 | 19,900 | | |
| 21 | **9b** (0.2) | MAO | 3000 | 25 | 19,500 | 9.46 | 1.56 |
| 22 | **9a** (0.2) | Et$_3$Al | 25 | 0 | 954 | 1.31 | 1.89 |
| 23 | **9a** (0.2) | $^i$Bu$_3$Al | 25 | 0 | 7430 | 1.43 | 1.72 |
| 24 | **9a** (0.2) | $^i$Bu$_3$Al | 50 | 0 | 6360 | 2.32 | 1.42 |
| 25 | **9a** (0.2) | $^i$Bu$_3$Al | 50 | 25 | 11,000 | 1.80 | 1.76 |
| 26 | **9a** (0.2) | Et$_2$AlCl | 2500 | 0 | 1970 | insoluble | |
| 27 | **9b** (0.1) | $^i$Bu$_3$Al | 600 | 25 | 22,400 | | |
| 28 | **9b** (0.02) | $^i$Bu$_3$Al | 2000 | 25 | 47,100 | | |
| 29 | **9b** (0.02) | $^i$Bu$_3$Al | 2500 | 25 | 66,000 | 13.7 | 2.35 |

$^a$ Conditions: ethylene 8 atm, toluene 30 mL. $^b$ Molar ratio to V. $^c$ Activity in kg-PE/mol-V·h. $^d$ GPC data in *o*-dichlorobenzene vs. polystyrene standards.

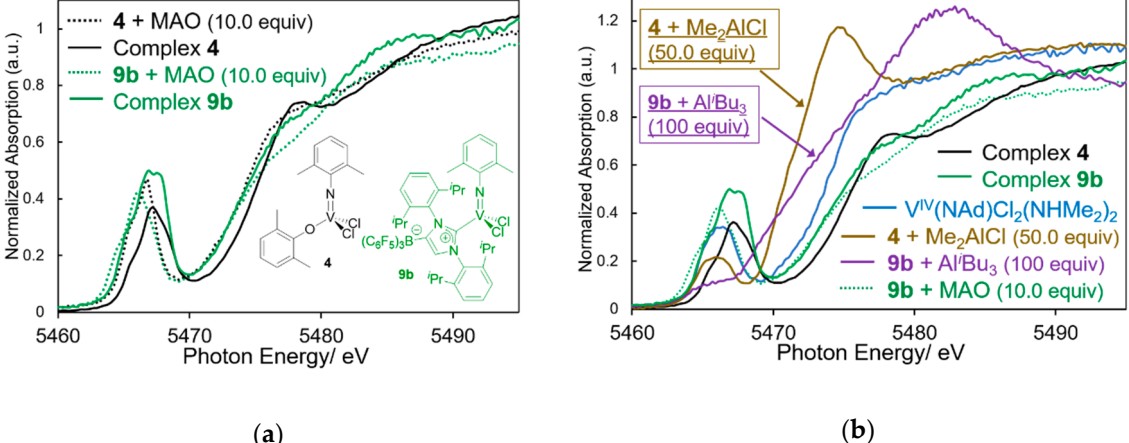

(**a**)            (**b**)

**Figure 5.** V K-edge XANES spectra (in toluene at 25 °C) for V(NAr)Cl$_2$(L) (L = OAr (**2**), WCA-NHC (**9b**)) with (**a**) 10 equiv of MAO [20,23], (**b**) 10.0 equiv of Et$_2$AlCl or 100 equiv of Al$^i$Bu$_3$ [23]. The spectrum of V$^{IV}$(NAd)Cl$_2$(NHMe$_2$)$_2$ is placed for reference.

In contrast, it should be noted that remarkable change in the edge peak (and the shoulder-edge at 5475.7 eV) was observed when **4** was treated with Me$_2$AlCl (50 equiv, Figure 5a) [23], and the spectrum did not change upon the further addition of NBE (50 equiv); similar spectral change was observed when **4** was treated with 50 equiv of Et$_2$AlCl [23]. The pre-edge peak in **4** (5466.8 (and 5465.3) eV) shifted to low energy or became one absorption band upon addition of Me$_2$AlCl (5465.7 eV). The observed edge absorption was apparently low compared to that in V(NAd)Cl$_2$(NHMe$_2$)$_2$, reference of vanadium(IV) complex [44]. The results thus strongly suggest that **4** was reduced by Me$_2$AlCl to afford certain vanadium(III) species [23].

Also note that significant change in the XANES spectrum was observed when **9b** was treated with Al$^i$Bu$_3$ (100 equiv, Figure 5b). Upon treatment of **9b** with Al$^i$Bu$_3$ the pre-edge absorption decreased and observed as a tiny shoulder at 5465.1 eV, clearly suggesting a structural change. Moreover, formation of vanadium(III) species by reduction was strongly suggested because of large low energy shift in the edge absorption [23]. The similar spectral change was observed when **9a** was treated with Al$^i$Bu$_3$ (100 equiv) [20]. Formation of vanadium(III) was also supported by $^{51}$V NMR spectra (disappearance of signal upon addition of Al$^i$Bu$_3$ due to formation of the paramagnetic species) and ESR spectra (negligible resonance in toluene solution consisting of **9a** and Al$^i$Bu$_3$) [20].

It should also be noted that the XANES spectra, the pre-edge intensities as well as the edge absorptions, in the reaction of **4** with Me$_2$AlCl or Et$_2$AlCl are apparently different from those in the

reaction of **9a,b** with Al$^i$Bu$_3$. Moreover, the pre-edge intensities in the reaction of **4** with Me$_2$AlCl are also different from that in the reaction of V(NAd)Cl$_2$(L) (**8**, L = 2-(2′-benzimidazolyl)-6-methylpyridine) with Me$_2$AlCl [22]. Further decrease in the pre-edge intensity was observed when ETA (Cl$_3$CCO$_2$Et) was added into the solution containing **8** and Me$_2$AlCl, whereas reaction of **4** with Me$_2$AlCl led to slight decrease in the intensity. These strongly suggest that different (imido)vanadium(III) species, which fold different geometry with coordinated ligands, play roles in the ethylene (co)polymerization.

Figure 6 shows V K-edge FT-EXAFS spectra (in toluene at 25 °C) for **4** with addition of Me$_2$AlCl (50 equiv), and **9a** with addition of Al$^i$Bu$_3$ (100 equiv), and Table 4 summarizes observed neighboring atoms and bond distances around vanadium on the basis of analysis [23]. It should be noted that the imido ligand in **4** was remained by reacting with Me$_2$AlCl (V–N, 1.64 ± 0.04 Å). Observed two V–Cl bonds (2.45 ± 0.03 Å) became apparently longer by treatment of **4** (V–Cl: 2.18 ± 0.03 Å) with Me$_2$AlCl, strongly suggesting that these Cl ligands could be neutral lone pair donors bridged through Al, expressed as V←:Cl-AlMe$_2$ or V←:Cl-Al(Cl)Me. Interestingly, the V–N and V–Cl bond distances did not change when **4** was treated with MAO, as observed in the EXAFS oscillation and the FT-EXAFS spectrum, whereas the coordination number of Cl decreased by treatment of **4** with MAO.

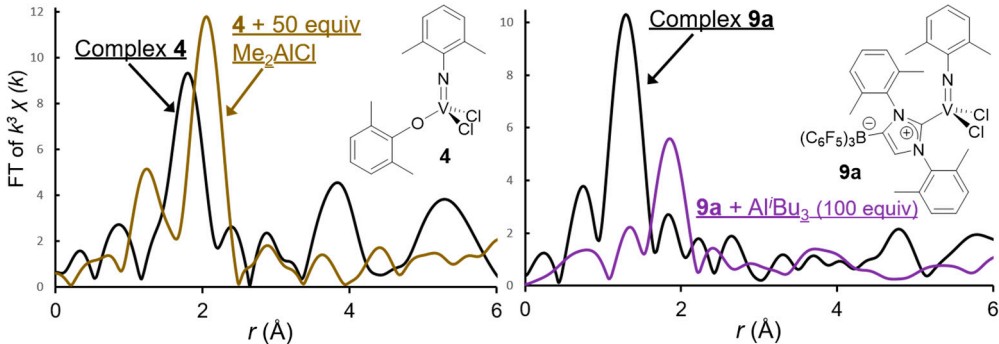

**Figure 6.** V K-edge FT-EXAFS spectra (in toluene at 25 °C) for reactions of (**left**; V(NAr)Cl$_2$(OAr); **4**, Ar = 2,6-Me$_2$C$_6$H$_3$) in the presence of Me$_2$AlCl (50 equiv), (**right**; V(NAr)Cl$_2$(WCA-NHC); **9a**) in the presence of Al$^i$Bu$_3$ (100 equiv)) [23].

**Table 4.** Summary of data for V(NAr)Cl$_2$(OAr) (**4**) in the presence of Me$_2$AlCl (50 equiv), or V(NAr)Cl$_2$(WCA-NHC-Ar) (**9a**) in the presence of Al$^i$Bu$_3$ (100 equiv) [23] [a].

| | Complex 4 | | 4 + Me$_2$AlCl | | | Complex 9a | | 9a + Al$^i$Bu$_3$ | |
| --- | --- | --- | --- | --- | --- | --- | --- | --- | --- |
| Atom | C.N. | *r* (Å) | C.N. | *r* (Å) | Atom | C.N. | *r* (Å) | C.N. | *r* (Å) |
| N(O) | 2.4 ± 0.3 | 1.80 ± 0.05 | 1.3 ± 0.2 | 1.64 ± 0.04 | N(C) | 2.1 ± 0.2 | 1.62 ± 0.03 | 0.8 ± 0.3 | 1.66 ± 0.17 |
| Cl | 1.9 ± 0.2 | 2.18 ± 0.03 | 2.0 ± 0.2 | 2.45 ± 0.03 | Cl | 1.0 ± 0.2 | 2.16 ± 0.04 | | |
| | | | | | Cl | 1.0 ± 0.2 | 2.34 ± 0.05 | 1.0 ± 0.2 | 2.34 ± 0.04 |

[a] Atom: neighbor atom, C.N.: coordination number, *r*: bond length.

The observed V–N distance by the EXAFS analysis (1.62 ± 0.03 Å) in **9a** is close to that determined by the X ray crystallography (1.654(3) Å). One of the observed V–Cl bond distance by the EXAFS analysis (2.16 ± 0.04 Å) was close to those determined by the X ray crystallography (2.1559(9) and 2.1620(10) Å), but another V–Cl bond distance (2.34 ± 0.05 Å) was, however, much longer than those in the related dichloride complexes (2.1901(8)–2.2462(8) Å, by X-ray crystallography) containing monodentate anionic donor ligands. Moreover, the V–C$_{carbene}$ bond (2.076(3) Å) was not observed or may be overlapped with the V–N bond (coordination number, 2.1 ± 0.2).

As shown in Figure 6 right, the FT-EXAFS spectrum of **9a** changed drastically by treating with Al$^i$Bu$_3$ (100 equiv, Figure 6 right), as observed in the XANES spectra of **9a,b**. As observed in **4**, the imido ligand in **9a** (V–N bond, 1.66 ± 0.17 Å) was preserved, and one V–Cl bond (2.34 ± 0.04 Å), which was longer than those in the reported (imido)vanadium(V) complexes with the anionic donor ligand

(shown above), was also observed. These analysis data however did not provide any information concerning the V–C$_{alkyl}$ bond, which should play a role for olefin insertions (polymerizations). In situ formation of (imido)vanadium(III)-alkyl species can be thus considered on the basis of NMR, ESR, and XANES spectra, therefore, as described above, Cl ligands could be neutral lone pair donors bridged through Al as suggested from the bond distances in the EXAFS analysis.

As shown in Figure 7a, remarkable changes in the XANES (pre-edge and edge region) spectrum were observed when V(NAd)Cl$_2$(L) (**8**, L = 2-(2'-benzimidazolyl)-6-methylpyridine) was treated with Me$_2$AlCl; the pre-edge peaks in 8 (5465.5 and 5467.7 eV) became one absorption upon addition of Me$_2$AlCl (5466.0 eV) [22]. Low energy shift in the edge absorption strongly suggests that **8** was reduced by reaction with Me$_2$AlCl accompanied with the structural change. The result is in good agreement with those in the $^{51}$V NMR spectrum (disappearance of signal due to generation of the paramagnetic species) and ESR spectra. Note that the intensity in the shoulder-edge (5475.7 eV) increased upon addition of ETA (Cl$_3$CCO$_2$Et) with decreasing the intensity of pre-edge peak (5465.5 eV); this corresponds to the fact that the activity increased upon addition of ETA. It is thus assumed that addition of ETA would be thus effective for generation (increased percentage) of catalytically active species [22].

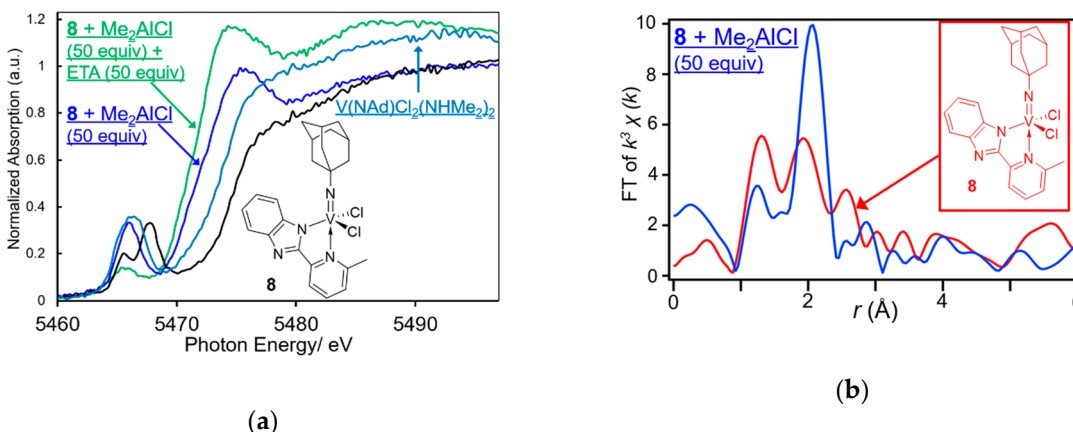

**Figure 7.** (**a**) V K-edge XANES spectra (in toluene at 25 °C) for V(NAr)Cl$_2$(L) (L = 2-(2'-benzimidazolyl)-6-methylpyridine (**8**)) with 50 equiv of Me$_2$AlCl, 50 equiv of Cl$_3$CCO$_2$Et (ETA). (**b**) FT-EXAFS spectra (in toluene at 25 °C) for reactions of **8** with 50 equiv of Me$_2$AlCl [22].

Figure 7b shows FT-EXAFS spectra and the analysis data are summarized in Table 5 [22]. Three V–N and two V–Cl bonds including the bond distances observed in the spectrum for **8** clearly suggest that the complex **8** preserves basic trigonal bipyramidal structure (determined by the X-ray crystallography) in solution. It turned out that the coordination number of the V–N bond decreased upon addition of Me$_2$AlCl, suggesting the ligand dissociation (probably L) with addition of Me$_2$AlCl. Presence of three (more than two) V–Cl bonds (2.455(7) Å), which became apparently weak compared to those in the original (2.293(3) Å) [22].

**Table 5.** Summary of data for V(NAr)Cl$_2$(L) (L = 2-(2'- benzimidazolyl)-6-methylpyridine (**8**)) in the presence of Me$_2$AlCl (50 equiv) [22] $^a$.

| | Complex 8 | | 8 + Me$_2$AlCl | |
|---|---|---|---|---|
| Atom | C.N. | r (Å) | C.N. | r (Å) |
| N | 1.7(2) | 1.683(5) | 0.9(3) | 1.64(2) |
| N | 1.2(8) | 2.290(42) | | |
| Cl | 1.6(2) | 2.293(3) | 2.6(1) | 2.455(7) |

$^a$ Atom: neighbor atom, C.N.: coordination number, r: bond length.

These analysis results clearly demonstrate that different (imido)vanadium(III) species, which contain a different number of V–Cl bonds (2.34–2.46 Å) as a neutral lone pair donor bridged through Al, play a role in these catalytic reactions; the species containing one V–Cl bond were formed from the reaction of **9a** with Al$^i$Bu$_3$, whereas the species containing two and three V–Cl bonds were formed from the reaction of **4**, V(NAd)Cl$_2$(L) (**8**, L = 2-(2′-benzimidazolyl)-6-methylpyridine) with Me$_2$AlCl, respectively. It is demonstrated that vanadium(III) species containing the imido V–N bond were formed by treatment with Me$_2$AlCl or Al$^i$Bu$_3$ in all cases. Although the V–C$_{alkyl}$ bond, which should play a role in the ethylene (co)polymerization, could not be defined through these studies, these results strongly suggest that three different active (Imido)vanadium(III) species were formed depending upon the anionic donor ligand and Al cocatalyst employed. In contrast, no significant changes in both XANES and EXAFS spectra were observed by reaction of these complexes with MAO, suggesting that the basic framework (a distorted 4 coordinate tetrahedral geometry) and the oxidation state were preserved; these results thus suggest that cationic (imido)vanadium(V) species play a role in the ethylene polymerization in the presence of MAO. These thus demonstrates that different catalytically active species containing imido ligand with different oxidation states play a role in ethylene (co)polymerization depending upon the anionic donor ligand and Al cocatalyst employed.

## 5. Concluding Remarks

The synchrotron XAS analysis is the important method for obtainment of the information not only the oxidation state and the basic structure (by XANES analysis of pre-edge and edge absorptions), but also the atoms coordinated to the active site (by FT-EXAFS analysis). The analysis in solids has been widely employed for studies in heterogeneous catalysis, but limited reports were known in homogeneous molecular catalysis. The solution XAS analysis of molecular complexes provides the structural and the electronic information, which are usually estimated on the data in the solid state (through X-ray crystallography). The approach should be thus very useful not only for better understanding in coordination chemistry and organometallic chemistry, but also for study in the catalysis mechanism. Moreover, we should also emphasize that, as shown in Figure 8, we did not need a specified facility for the measurement except a certain synchrotron facility (e.g., SPring-8, BL01B1 beamline); it was possible to prepare the samples on site in the drybox. The method can be also useful for analysis of titanium catalysts for olefin polymerization [86].

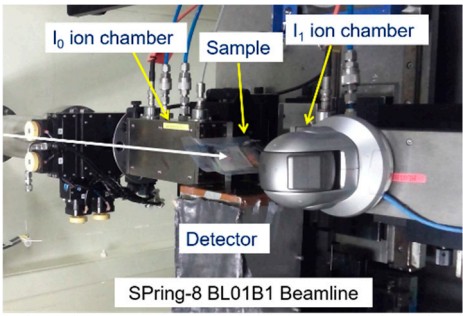

**Figure 8.** Analysis apparatus for measurement of solution V K Edge XANES and EXAFS spectra (5.46 keV, through synchrotron radiation at SPring-8, BL01B1 beamline in toluene at 25 °C).

In this feature article, study on the solution XAS analysis of the catalytically active species in ethylene dimerization and polymerization using molecular vanadium catalysts that contain both imido and anionic donor ligands in the presence of Al cocatalysts was introduced. In particular, formations of certain vanadium(III) species, which could not be observed by ESR nor $^{51}$V NMR spectra, were clearly demonstrated by reduction with Al$^i$Bu$_3$ or Me$_2$AlCl, which were proposed to be the active species, by adopting this analysis method. As described in the introductory, this was, as far as we believe, the first demonstration of observation/analysis of (NMR and ESR silent) vanadium(III) species applied to study

in homogeneous catalysis by using solution XANES and EXAFS analysis. Moreover, it turned out that different (imido)vanadium(III) species were generated by reactions of (imido)vanadium(V) complexes containing different anionic donor ligands with Al alkyls through the analyses. The information through this study should be potentially important for better understanding. The solution XAS analysis data in addition to the NMR and ESR spectra as conventional methods (and crystallographic analysis and reaction chemistry) should contribute to provide clear information for study in catalysis, organometallic chemistry. Therefore, the author highly hopes that more research papers will be seen in the next decade in the field of homogeneous catalysis and organometallic chemistry.

**Funding:** This project was partly supported by Grant-in-Aid for Scientific Research on Innovative Areas ("3D Active-Site Science", No. 26105003) from The Ministry of Education, Culture, Sports, Science and Technology (MEXT), Japan, and Grant-in-Aid for Scientific Research from the Japan Society for the Promotion of Science (JSPS, No. 15H03812, 18H01982), and Fund for the Promotion of Joint International Research (Fostering Joint International Research, 19KK0139).

**Acknowledgments:** The author expresses his deeply heartfelt thanks to S. Yamazoe (Tokyo Metropolitan University, TMU), T. Mitsudome (Osaka University), and T. Ina (Japan Synchrotron Radiation Research Institute, JASRI) for big support in XAS analysis. The author also thanks to K. Tsutsumi, A. Igarashi, H. Hayashibara, G. Nagai, T. Omiya, H. Harakawa, M. Kuboki, K. Inoue, I. Izawa, K. Kawamura and Y. Kawamoto (TMU) for technical assistance for the XAS analysis at the BL01B1 beam line at the SPring-8 (JASRI, proposal no. 2015B1308, 2016A1455, 2016B1509, 2017A1512, 2018A1245, 2018B1335). The author thanks Tosoh Finechem Co. for donating MAO.

**Conflicts of Interest:** The authors declare no conflict of interest.

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
