# Peer review of "Solution X-Ray Absorption Spectroscopy (XAS) for Analysis of Catalytically Active Species in Reactions with Ethylene by Homogeneous (Imido)vanadium(V) Complexes—Al Cocatalyst Systems"

_catalysts, doi:10.3390/catal9121016_

Round 1

Reviewer 1 Report

The paper by Nomura is a review article which describes the characterization of the catalytically active vanadium(V) species and intermediates in reactions with ethylene using aluminum complexes as co-catalyst. The area is rather narrow and specific and the value of this review article for a broader group of chemists is doubtful. The data quality of the experimental data put forward as examples a rather low. The conclusions are in most cases drawn from XANES spectra. The interpretation of XANES are more uncertain than of full EXAFS data. To improve analysis of the XANES data the author could use advanced programs for the analysis of XANES spectra as MXAN, but this is not the case here. In order to improve an article of this kind time resolved studies should have been performed as has been demonstrated in a number of articles in recent years. As it is presented here it is snapshots of reactions in solution using XANES without the possibility to follow possible structural changes around vanadium. I regard this manuscript more of a summary of the authors own work more than a true review article of a field relevant for wide group catalyst scientists. The English needs to be improved significantly.

In my opinion, this paper does not keep the standards required for a review article in general or for a paper in Catalysts. I therefore recommend rejection of this manuscript.

Author Response

Dear reviewer 1:

Thank you for your critical comment.  As you may know, recent advances in analysis of this field including detector enables us to obtain more concise information concerning the catalytically active species, and this article is, as you commented, a feature article rather than mini review including basics in this analysis.  In recent decades, as you may know, many researchers in the molecular catalysis (organometallics) have an interest in this approach, and I thus highly believe that this article will be a good introduction for the uninitiated researchers (since I placed a picture in the analysis).  We have also demonstrated several EXAFS analysis data concerning the role of anionic donor ligand toward the structure of the active (or dormant) species in the catalysis solution.

As we demonstrated very recently (ASAP in Organometallics), the approach is not only useful in vanadium catalysis, but also in mechanistic study in early transition metal catalysis especially with high oxidation state. 

I thus highly believe this is a good occasion to submit this reviewing article by invitation, and I hope you kindly understand this situation.

Yours sincerely,

Kotohiro Nomura

Reviewer 2 Report

This review is focused on the utilization of different types of synchrotron-based X-ray absorption spectroscopies to get the mechanistic information on ethylene polymerization catalyzed by various vanadium complexes. Such investigation, non-trivial in general, is additionally complicated in this case by paramagnetic nature of V(III) species directly involved in the process, which cannot be observed by ESR spectroscopy. The use of these X-ray spectroscopic methods in solution becomes more and more popular during last two decades due to easier access to highly intense X-ray sources, and it is important to accumulate the reliable data for further development of these techniques, even if the currently obtained results are not extremely informative and difficult to analyze.

The manuscript is well written and in my opinion deserves publication. However, one technical issue concerning the graphical presentation of vanadium complexes 9a-9b has to be fixed to remove the ambiguity. Indeed, the NHC ligand is always two-electron donor (coordination flash in the style used by the authors) and not single electron X ligand with positive charge in the imidazolium moiety. The most correct modern representation of such anionic imidazol-2,4-diylidene L2-type ligands (see for example Coord. Chem. Rev. 2015, 293 – 294, 80; Coord. Chem. Rev. 2016, 316, 68) will be the use of two coordination flashes to V and B fragments, the delocalized anionic charge in the NHC moiety and cationic charge at the metal atom required for V3+ configuration.

Author Response

Dear reviewer 2:

Thank you for your critical comment on drawing for 9a,b.  As you may find in the original manuscript, we often discussed with this matter with the coauthor (author in the Coord. Chem. Rev. introduced in the above).  On the basis of X-ray crystallographic analysis (V-C bond distance), it seems likely that the cationic charge would be delocalized in the NHC.  We thus draw the structure in the present form.  However, in vanadium(III) species, it seems likely that NHC would play a role as a neutral donor as you commented (although we did not place any in the manuscript), and we are now explore the details through the EXAFS analysis.  I hope you kindly understand this matter.

Yours sincerely,

Kotohiro Nomura

Reviewer 3 Report

This paper deals with the spectroscopic characterization of various imidovanadium(V) homogeneous catalysts for the ethylene polymerization. These complexes have been studied by XAS spectroscopy in solution, including XANES and EXAFS analyses. The characterization of these compounds has been performed in the presence of several aluminium-based co-catalysts in order to determine the true nature of the catalysts. It is noteworthy that this manuscript also contains a substantial introduction to the XAS of vanadium complexes, so that this manuscript may be considered as a reviewing feature article.

To my opinion, the reviewing part is of suitable size, well-documented and particularly interesting, with chosen examples that allowed the readers to understand quite easily the demonstration proposed in the second part of manuscript. In general, the results presented in this second part are clearly and meticulously exposed, in order to draw an unambiguous picture of the catalyst behaviour in solution before the addition of the substrate.

            A few comments: please clarify the nature of the VIV oxide used in paragraph 2,VO2, sometimes defined as “V2O4”, which is confusing. Furthermore, there are different types of VO2 structures, with different vanadium geometries. If I understand well, one of the different monoclinic phases has been used as models in this study.

            Finally, I do not understand the concept of chloride defined as “neutral (or weak anionic) ligand”. Chloride being necessarily mono-anionic, please clarify what is implied in this wording.

In conclusion, this manuscript is a nice piece of work, which may be published in high standard journals. I recommend the present work for publication as a full paper in “Catalysts” in its actual state provided the authors meet the few comments listed above.

Author Response

Dear reviewer 3:

Thank you for your critical comment.  Chloride play a role as anionic donor or lone pair donor (as described in Scheme 2, bottom) through Al in this case.  Therefore, number of Cl neutral donor ligand (longer than V-Cl sigma bond) in vanadium(III) species are dependent upon the starting vanadium(V) complexes (nature of anionic donor ligand) employed.  Additional explanation has been placed in the revised manuscript for better understanding.  I hope this would fulfill your request.  Concerning the vanadium oxide, we confirmed that the sample was VO2, and the text has also been revised.

Yours sincerely,

Kotohiro Nomura

Reviewer 4 Report

The paper/review by Nomura is very interesting and the method described (solution X-ray absorption Spectroscopy-XAS) for the analysis of the nature of the active species active in the polymerization and oligomerization of ethylene with homogeneous catalytic systems based on vanadium complexes in particular, but also on other homogeneous transition metal complexes in general, appears to be very powerful, accompanied also to the usual solution NMR analysis and RX analysis of the complexes in the solid state.

It seems to me that the text is quite clear and understandable, and that therefore the manuscript can be accepted for publication in the present form.

Author Response

Dear reviewer 4:

Thank you for your positive comment concerning the manuscript. I have revised the manuscript according to the reviewer's comments and the comments from the editorial office.  I believe this would be fine.

Yours sincerely,

Kotohiro Nomura